# Uncertainty-Aware Pseudo-labeling for Quantum Calculations

**Kexin Huang**[1,2]        **Vishnu Sresht**[1]        **Brajesh Rai**[1]        **Mykola Bordyuh**[1]

[1]Machine Learning and Computational Sciences, Pfizer Inc., Cambridge, Massachusetts, USA
[2]Department of Computer Science, Stanford University, Stanford, California, USA

## Abstract

Machine learning models have recently shown promise in predicting molecular quantum chemical properties. However, the path to real-life adoption requires (1) learning under low-resource constraints and (2) out-of-distribution generalization to unseen, structurally diverse molecules. We observe that these two challenges can be addressed via abundant labels, which is often not the case in quantum chemistry. We hypothesize that pseudo-labeling on a vast array of unlabeled molecules can serve as gold-label proxies to expand the training labeled dataset significantly. The challenge in pseudo-labeling is to prevent the bad pseudo-labels from biasing the model. Motivated by the entropy minimization framework, we develop a simple and effective strategy PSEUD$\sigma$ that can assign pseudo-labels, detect bad pseudo-labels through evidential uncertainty, and prevent them from biasing the model using adaptive weighting. Empirically, PSEUD$\sigma$ improves quantum calculations accuracy in full data, low data, and out-of-distribution settings.

## 1 INTRODUCTION

Ab initio quantum chemistry methods attempt to solve the electronic many-body Schrödinger equation to characterize biomolecular properties and interactions at different level of theory and numerical approximations. Despite extensive repertoire of methods from Post–Hartree–Fock methods such as CCSD(T) (coupled cluster single-double-triple) and MP2 (second order Møller-Plesset) [Watts et al., 1992] to Density Functional Theory (DFT) [Parr and Weitao, 1989] they continue to be numerically expensive, even with recent advances in hardware capabilities. Machine learning (ML) models have astonishing performance in approximat-

ing these calculations at a fraction of the computational cost [von Lilienfeld and Burke, 2020]. Such speedups have the potential to accelerate the discovery of new materials and therapeutics.

Most publications on this topic have relied on QM9 dataset, a standard benchmark for training and evaluating ML models to predict QM properties of small molecules precomputed using approximate DFT calculations. Model-centric approaches demonstrated great capabilities of the machine learning on this dataset, by showing low error on hold-out test set of unseen molecules (e.g. [Schütt et al., 2017, Klicpera et al., 2020, Liu et al., 2021]). Despite the promise, realistic adoptions still face unsolved challenges. First, previous ML models rely on large number of labeled molecular geometries (e.g. 100K for QM9), which are often not available for higher-fidelity level of energy calculations such as CCSD(T) or MP2 – the challenge is for a QM/ML model to perform well under small number of computed geometries; Second, previous works evaluate the trained ML models on a test set that is in a similar chemical space as the training set (i.e. in-distribution), while the goal of deployment is to predict energies for structurally distant molecules across the diverse chemical space – the challenge is for a QM/ML model to generalize to out-of-distribution molecules. We observe that known QM/ML architectures would have significantly higher errors in these difficult regimes, calling for innovative ML algorithms to tackle these challenges (Section 6).

**Present work.** Our work focuses on addressing the fundamental cause of the above challenges - the scarcity of computed QM labels on a diverse set of chemicals. We utilize the abundance of the unlabeled molecules and develop an effective pseudo-labeling strategy suitable for QM calculations. The basic idea behind pseudo - labeling is to estimate the labels for the unlabeled data and expand the training dataset. Several pseudo-labeling methods in various machine learning domains have been already applied successfully to improve state-of the-art models especially

*Accepted for the 38th Conference on Uncertainty in Artificial Intelligence* (UAI 2022).

in computer vision [Xie et al., 2020, Lee et al., 2013, Iscen et al., 2019]. We investigate many of these approaches (e.g., data augmentation, model noise, student-training, re-initialization) and found many can have a negative effect in quantum calculations. For example, adding positional noise [Xie et al., 2020] in molecular geometries could significantly affect energies and thus bias the pseudo-labels. Thus, a QM-specialized pseudo-labeling strategy is needed. After extensive empirical studies, we reached an optimal QM-specialized scheduling strategy using episodes with no re-initialization and noise.

A crucial issue in pseudo-labeling is the introduced bias from low-quality pseudo-labels. Based on theoretical motivations (Section 5), we rely on a key observation that a data point with less evidence/higher model uncertainty is more likely to be of low-quality pseudo-label (Section 6). Thus, we use model-generated evidential uncertainty to quantify each unlabeled data and use it to adaptively lower the weights of bad pseudo-labels in the training loss to reduce the bias effect.

In summary, our method focuses on the effective strategy to incorporate QM pseudo-labels to alleviate the fundamental label scarcity issue, along with the associated challenges of low-data and out-of-distribution generalization. We have made the following contributions: (1) Previous QM/ML methods focus on in-distribution and label abundant setting while we investigate more realistic case of low-data and out-of-distribution settings; (2) Pivoting away from the status quo in improving the physics-based representation, we propose to look at data-centric approaches on learning from the vast array of unlabeled molecules; (3) We propose a simple, effective, theoretically motivated pseudo-labeling strategy PSEUDσ designed specifically for QM, integrating episodic scheduling and downplaying low-quality pseudo-labels informed by uncertainty; (4) Empirically, we show that PSEUDσ can improve QM accuracy for any atomistic model across full-data, low-data, and out-of-distribution settings.

## 2 RELATED WORKS

**ML-aided quantum calculations.** Recently, many ML models have been proposed to improve quantum calculations. They mainly focus on improving the physics-based representation and architectural developments tested on the full QM9 dataset [Schütt et al., 2017, Unke and Meuwly, 2019, Anderson et al., 2019, Lu et al., 2019, Klicpera et al., 2020, Liu et al., 2021, Qiao et al., 2021]. In contrast, our work proposes to shift the focus to model-agnostic training strategies in realistic low-data and out-of-distribution settings.

**Pseudo-labeling.** Pseudo-labeling/self-training generates pseudo-labels for unlabeled data. Numerous works on how

to assign pseudo-labels exist, notably, through trained ML model predictions [Lee et al., 2013], label propagation[Shi et al., 2018, Iscen et al., 2019], and history cache [Likhomanenko et al., 2021, Higuchi et al., 2021]. PSEUDσ is different as it focuses on detecting and preventing bad pseudo-labels from affecting the model. Also, PSEUDσ adopts a novel episodic pseudo-labelling strategy with a re-initialized learning rate. [Xie et al., 2020] re-initialize the network as a student when a new pseudo-label set is generated along with noise per epoch. In contrast, PSEUDσ has no student and no noise as both are shown to be ineffective for QM in Section 6. In addition, small perturbational noise in 3D molecular geometry could easily lead to a drastic energy change. Thus, a naive strategy of adding noise does not work for QM tasks. More related is a preceding work [Rizve et al., 2021] that develops an uncertainty-aware pseudo-labeling strategy. They introduce additional hyperparameters to remove pseudo-labels at some uncertainties. In contrast, PSEUDσ uses an effective adaptive weighting scheme, along with an episodic pseudo-labeling training schedule. There are also works using adaptive weighting scheme to leverage unlabeled data. [Ren et al., 2020] adopt a hessian based approach for weighting, while we utilize an uncertainty based weighting based on a theoretical motivation. Additionally, PSEUDσ is the first method that studies pseudo-label in quantum calculations that present unique challenges.

**Uncertainty.** Model uncertainty is a well-studied subject [Kendall and Gal, 2017, Lakshminarayanan et al., 2017, Blundell et al., 2015]. Notably, [Berthelot et al., 2020] estimate marginal class distribution for consistency regularization. Their work can also be connected to the entropy minimization strategy. [Lienen and Hüllermeier, 2021] use a credal set to extend to the use of multiple probability distributions to reduce the bias of pseudo-labels. In contrast, we adopt a different approach - explicit uncertainty modeling (evidential modelling for capturing data and model uncertainty) for a different goal (measuring pseudo-label quality for adaptive reweighting). More related to us, [Amini et al., 2020] use evidential uncertainty to add a prior over the gaussian parameters to search for higher-order patterns for regression tasks. PSEUDσ leverages evidential uncertainty as the uncertainty measure. Note that PSEUDσ is uncertainty measure-agnostic. We can easily switch to alternative uncertainty measures. Recently, [Soleimany et al., 2021] adapt evidential uncertainty and show that it can successfully help guide property prediction. In contrast, we leverage evidential uncertainty as a proxy for pseudo-label quality to tackle low-data and out-of-distribution challenges in a realistic quantum calculations setup.

## 3 PROBLEM FORMULATION

Let $\mathcal{X} = \{\mathbf{x}_1, \dots, \mathbf{x}_N\}$ denote $N$ molecules, where each molecule $\mathbf{x}_i$ is uniquely defined by 3D coordinates

$\{(a_j^i, b_j^i, c_j^i)\}_{j=1}^{N_i}$ for $N_i$ atoms with atom types $\{t_j\}_{j=1}^{N_i}$ in the corresponding molecule. We then denote $\mathcal{Y} = \{y_1, \ldots y_N\}$ a set of quantum mechanical properties for each molecule, where $i$-th molecule has label $y_i$. The labeled dataset thus consists of a set of pairs of 3D coordinates and scalar labels $\mathcal{D} = \{\mathcal{X}, \mathcal{Y}\}$.

In addition to the labeled data, we solicit a large quantity of unlabeled data to generate pseudo-labels. We denote an unlabeled dataset $\mathcal{U} = \{\mathbf{x}_1, \ldots, \mathbf{x}_P\}$, where $P$ is the size of the unlabeled dataset. Given an atomistic model $f(\cdot)$, we can generate pseudo-labels $\hat{\mathcal{Y}} = \{\hat{y}_1, \ldots, \hat{y}_P\}$, where $\hat{y}_i = f(\mathbf{x}_i)$ for $\mathbf{x}_i \in \mathcal{U}$.

The problem is to train a machine learning-based atomistic model $f : \mathbf{x} \mapsto y$ that can establish an accurate map from 3D coordinates to the quantum mechanical properties of the molecules with the help of pseudo-labeled dataset $\mathcal{U}$.

# 4 PSEUD$\sigma$: UNCERTAINTY-AWARE PSEUDO-LABELING FOR QUANTUM CALCULATIONS

PSEUD$\sigma$ (Figure 1) is an approach for quantum chemical property prediction. Building on theoretical motivation from Section 5, PSEUD$\sigma$ solicits pseudo-labels on a vast array of an unlabeled dataset to increase the diversity of the training space via an episodic labeling strategy. Then, it adaptively weights the pseudo-labels using evidential uncertainty to allow a positive transfer. The overview is in Algorithm 1.

**Episodic Pseudo-labeling.** We devise a pseudo-labeling strategy that can ensure learning from the pseudo-labels to the fullest extent for QM. We have made two distinct modifications compared to existing works. First is the pseudo-label scheduling. In the standard pseudo-labeling [Lee et al., 2013], pseudo-labels are updated in every update and the model is continuously trained. In contrast, we devise an episodic training strategy, where each episode consists of $K$ epochs, and pseudo-labels are regenerated in every episode, while the model is continuously trained. This is important because we observe that updating pseudo labels too frequently prevents the model from extracting all the useful information from pseudo-labels. In contrast, our episodic approach gives the model more time to absorb useful information from a given set of pseudo-labels. A second modification is how we carry out model updates. In self-training [Rizve et al., 2021, Xie et al., 2020], a set of pseudo-labels are regenerated after $K$ epochs (1 episode) and the model is reinitialized. Instead, we train the same model across episodes. This new strategy allows the model to be exposed to a larger number of labels or training data points given the same time frame. For each episode, we also reinitialize the learning rate with a small step-wise decay strategy to allow the model a chance to jump out of the local optimum from the previous set of pseudo-labels.

Formally, PSEUD$\sigma$ mainly consists of three stages: in the first stage, regular training is conducted on labeled data $\mathcal{D}$, and the output model is the initialized model $f^{(1)}$. In the second stage, the updated model at episode $k$ then conducts inference on the entire unlabeled data $\hat{\mathcal{Y}} = f^{(k)}(\mathcal{U})$ to generate the pseudo-label set. The per-episode pseudo-label set is then combined with the gold-labeled data to form the training data for the next episode. In the third stage, the model is further trained using the combined dataset to get a new model $f^{(k+1)}$ after $K$ epochs (i.e., one more episode). The second and third stages are then reiterated till the loss converges.

**Evidential uncertainty quantification.** Pseudo-labels are noisy. Many are incorrect and can potentially lead to negative transfer. Thus, it is instrumental to determine the quality of pseudo-labels. However, there is no auxiliary information in the dataset about the pseudo-labels. Thus, we need to quantify it through some proxies that can be assigned without auxiliary information. Our key observation is that low-quality pseudo-labels have high model uncertainty, and high-quality pseudo-labels have low model uncertainty. Another advantage of model uncertainty is that it can be estimated solely from $\mathbf{x}$, if we make it model uncertainty-aware.

Building on the theoretical motivation about the connection between evidential uncertainty and the entropy minimization in Section 5, we use evidential uncertainty as the proxy for label quality. The evidential modeling of molecular property allows us to derive an analytical expression for uncertainty, which can be directly used to weight the pseudo-labels. Formally, we can model the label probabilistically as being drawn from $(y_1, \cdots, y_i) \sim \mathcal{N}(\mu, \sigma^2)$, where the mean $\mu$ are variance $\sigma^2$ are unknown. To estimate them, we pose a prior

$$\mu \sim \mathcal{N}(\gamma, \sigma^2 v^{-1}), \sigma^2 \sim \Gamma^{-1}(\alpha, \beta), \tag{1}$$

where the parameters $\theta = (\mu, \sigma)$ is an instantiation of the posterior $p(\mu, \sigma^2 | \gamma, v, \alpha, \beta)$. The choice of prior allows the factorization $p(\mu, \sigma^2) = p(\mu) p(\sigma^2)$ [Jordan, 2009]. The posterior then becomes a $\mathrm{NormalInvGamma}(\gamma, v, \alpha, \beta)$ where the maximum likelihood estimation of $\theta$ can be analytically found as

$$\mathbb{E}[\mu] = \gamma, \quad \mathbb{E}[\sigma^2] = \frac{\beta}{\alpha - 1}. \tag{2}$$

Here, $\mathbb{E}[\sigma^2]$ plays the role of the aleatoric (data) uncertainty. The uncertainty of the model prediction can also be calculated, i.e. epistemic uncertainty:

$$\mathrm{Var}[\mu] = \mathbb{E}[\sigma^2]/v = \frac{\beta}{v(\alpha - 1)}. \tag{3}$$

As the MLE is deterministic, the model can output four prior parameters $\{\gamma, v, \alpha, \beta\}$ directly where the prediction and uncertainty can be derived from them analytically. The prior

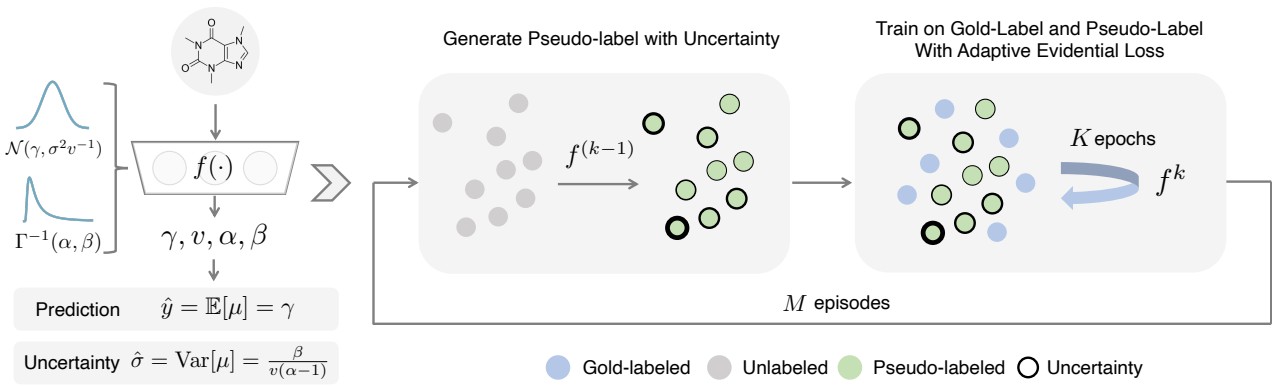

Figure 1: PSEUDσ illustration. In every episode $k$, PSEUDσ assigns pseudo-labels along with their evidential uncertainty using trained neural network $f^{(k-1)}$ from previous episode. The uncertainty is used as weight to adaptively adjust the loss in this episode's neural network $f^{(k)}$'s training to reduce the effect of bad pseudo-labels in an inner-loop training with $K$ epochs.

is optimized by evidential loss $\mathcal{L}^{\text{evi}}$ [Amini et al., 2020]:

$$\mathcal{L}_i^{\text{evi}} = -\log \text{St}\left(y_i; \gamma, \frac{\beta(1+v)}{v\alpha}, 2\alpha\right) + \lambda|y_i - \gamma|(2v+\alpha),$$
(4)

where the first term maximizes the log-likelihood of the posterior predictive, which is the Student's t-distribution. The second term is a regularizer that imposes a penalty whenever there is an error in the prediction and scales with the total evidence $2v + \alpha$ of our inferred posterior. Similarly, it encourages lower uncertainty when the model prediction is error free. This encourages the model to generate an accurate estimate of uncertainty or the degree of errors for the pseudo-labeled data points. The regularization is controlled by a hyperparameter $\lambda$.

**Adaptive weighting.** The evidential uncertainty detects the low-quality pseudo-labels. The next step is to remove the noisy effect from the model training. Naive methods often use removal based on a threshold [Rizve et al., 2021]. However, this has two disadvantages: (1) it introduces a new hyperparameter - the threshold; and (2) it removes a portion of unlabeled noisy data, which can contain useful information. Instead, we propose an adaptive weighting mechanism that weights the evidential loss with the inverse of the epistemic uncertainty. Intuitively, a higher uncertainty data point should have a smaller effect on the loss function because it is more likely that the sample has a low pseudo-label quality, and we want to reduce its effect on the model. Conversely, if a pseudo-label has low uncertainty, the label quality is high enough to be used as a high-fidelity proxy for a gold-label. Thus, it should have a higher impact on the loss. The uncertainty is from the teacher model in the previous episode and is fixed throughout the current episode. Thus, the adaptive weight for each pseudo data point $i$ becomes

$\hat{\mathcal{W}}_i = \text{Var}[\mu]_i^{-1}$. The final loss then becomes

$$\mathcal{L} = \frac{1}{|\mathcal{D}|}\sum_{i\in\mathcal{D}}\mathcal{L}_i^{\text{evi}} + \sum_{i\in\mathcal{U}}\frac{\hat{\mathcal{W}}_i}{\sum_{i\in\mathcal{U}}\hat{\mathcal{W}}_i}\mathcal{L}_i^{\text{evi}},$$
(5)

where the first term, corresponding to the labeled dataset $\mathcal{D}$, does not have weights unlike the second term corresponding to the unlabeled data $\mathcal{U}$. This adaptive loss solves two disadvantages: it has zero hyperparameters, and it removes the effect of bad pseudo-labels while retaining all training examples including the noisy ones to maximize the diversity of the training space.

## 5  PSEUDσ MOTIVATION: CONNECTION TO ENTROPY MINIMIZATION

We derive motivation about why evidential uncertainty and the weighting mechanism could be beneficial to pseudo-labeling based on the entropy minimization framework for semi-supervised learning from Grandvalet and Bengio [2004, 2006], Lee et al. [2013]. Notably, our use of Bayesian modeling enables us to analytically derive a conditional entropy for pseudo-labeled data. We find that evidential loss strongly relates to conditional entropy, and minimizing evidential loss directly minimizes entropy. Secondly, we find the conditional entropy could be decomposed into the inverse epistemic uncertainty and the log-likelihood, which motivates our weighting mechanism.

In the standard regression setting, one seeks to maximize the likelihood of the model $p_\theta(\mathcal{Y}|\mathcal{X})$ on the labeled data set $\mathcal{D}$. To utilize the unlabeled data set, we need to extract some useful information on how the model behaves on the unlabeled dataset and inject this information to improve the model. To measure the utility, entropy $\mathcal{H}(Y|\mathcal{U})$ is

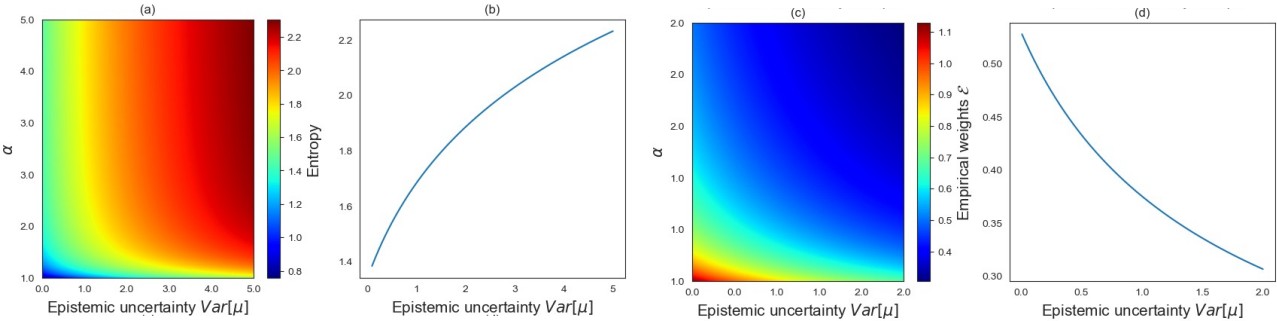

Figure 2: (a) Dependence of the entropy (Eq. 9) on epistemic uncertainty and virtual observation parameter $\alpha$ for a fixed aleatoric uncertainty $\mathbb{E}[\sigma^2] = 1$. As the the epistemic uncertainty increases, the entropy is also increases for all values of parameter $\alpha$. For example, figure (b) demonstrate the trend for a fixed $\alpha = 2$. Figure (c) demonstrates the dependence of empirical weights (Eq. 14) on epistemic uncertainty. The empirical weights tend to decrease as the epistemic uncertainty increases. Figure (d) demonstrates this trend for a fixed $\alpha = 2$.

introduced [Grandvalet and Bengio, 2006] as a proxy to measure the amount of information in unlabeled data:

$$\mathcal{H}(\mathcal{Y}\,|\,\mathcal{U}) = \sum_{\mathbf{x}_i \in \mathcal{U}} \mathbb{E}_{y \sim \mathrm{p}_\theta(y\,|\,\mathbf{x}_i)}[-\log \mathrm{p}_\theta(y\,|\,\mathbf{x}_i)]. \quad (6)$$

Throughout the text, we are referring to entropy as Shannon entropy. High entropy is associated with random predictions while low entropy is associated with non-random behavior. Hence, we hypothesize that small entropy may be indication of a signal that our model can benefit from. Small entropy, as seen below, corresponds to high model confidence and vice versa. Large entropy corresponds to high model uncertainty. Entropy minimization framework casts the regression as the following optimization problem:

$$\mathrm{argmax}_\theta \left[\log \mathrm{p}_\theta(\mathcal{Y}|\mathcal{X}) - c\,\mathcal{H}(\mathcal{Y}\,|\,\mathcal{U})\right], \quad (7)$$

where $c$ is the proportionality constant. Intuitively, here, the objective tends to maximize the log-likelihood on the labeled dataset while minimizing the entropy on the unlabeled data set at the same time to transfer knowledge from unlabeled data.

In previous works [Schütt et al., 2017, Liu et al., 2021], molecule properties are not modeled probabilistically such that entropy calculation is infeasible. In contrast, PSEUD$\sigma$ uses Bayesian modeling approaches that allow us to analytically calculate the entropy. For every molecule $\mathbf{x}_i$ the machine learning model outputs four parameters $f(\mathbf{x}_i) = (\alpha_i, \beta_i, \gamma_i, \nu_i)$. Based on these parameters, the likelihood of label $y$ given the input molecule $\mathbf{x}_i$ is given by the Student's t-distribution in the context of evidential regression

$$p_\theta(y|\mathbf{x}_i) = \mathrm{St}(y; \gamma_i, \sigma^2_{st,i}, 2\alpha_i) \quad (8)$$

evaluated at location parameter $\gamma_i$, Student's t-distribution scale parameter $\sigma^2_{st,i} = \frac{\beta_i(1+\nu_i)}{\nu_i \alpha_i}$ and $2\alpha_i$ degrees of freedom. The entropy of the Student's t-distribution given in

terms of evidential parameters is readily available (Appendix 1):

$$\mathcal{H}(y\,|\,\mathbf{x}_i) = \frac{2\alpha_i + 1}{2} \left( \Psi(\frac{2\alpha_i + 1}{2}) - \Psi(\alpha_i) \right)$$
$$+ \log \sqrt{2\alpha_i}\, \mathrm{B}(\alpha_i, \frac{1}{2}) + \frac{1}{2}\log \sigma^2_{st,i}, \quad (9)$$

where $\Psi$ is a digamma function and $\mathrm{B}(\cdot, \cdot)$ is a beta function. If we take model (epistemic) uncertainty (Eq. 3) as our measure of uncertainty, we can show that minimizing entropy directly relates to minimizing epistemic uncertainty. We plot the relation between the entropy and epistemic uncertainty in the Figure 2.

As the next step, we aim to uncover the dependence of the entropy on the model uncertainty of our pseudo-labeling approach. This can be done if we make two simplifications in entropy evaluation. Firstly, to introduce iterations as in pseudo-labeling, we replace entropy with the cross-entropy between two probability distributions: the predictions $y$ are generated from the probability distribution $p_{\theta(t-1)}(y|\mathbf{x}_i)$ at iteration step $t-1$ and log-likelihood are evaluated with respect to probability distribution $p_{\theta(t)}(y|\mathbf{x}_i)$ at iterative step t:

$$\mathcal{H}(\mathcal{Y}\,|\,\mathcal{U}) \approx \sum_{\mathbf{x}_i \in \mathcal{U}} \mathbb{E}_{y \sim p_{\theta(t-1)}(y|\mathbf{x}_i)} \left[ -\log \mathrm{p}_{\theta(t)}(y\,|\,\mathbf{x}_i) \right]. \quad (10)$$

Upon convergence, $t \to +\infty$, the probability distributions at every iterative step $p_{\theta(t-1)}(y|\mathbf{x}_i) \approx p_{\theta(t)}(y|\mathbf{x}_i)$ and are approximately the same and one can view introduced cross-entropy with respect to time step $t$ as entropy. At the earlier stages of training, cross-entropy acts as a regularizer encouraging network parameters $\theta(t)$ to match $\theta(t-1)$.

As a second approximation, to uncover model uncertainty in mathematical formulas, we approximate the probability distribution at time step $t-1$. We resort to empirical estimate of the entropy, as done in [Grandvalet and Bengio,

**Algorithm 1:** PSEUD$\sigma$ Algorithm.

---

**Input**: Labeled data $\mathcal{D} = \{(\mathbf{x}_1, y_1), \cdots, (\mathbf{x}_N, y_N)\}$,
unlabeled data $\mathcal{U} = \{\mathbf{x}_1, \cdots, \mathbf{x}_P\}$ $\hat{\mathcal{U}} \leftarrow \{\}, \hat{\mathcal{W}} \leftarrow \{\}$
`// Initialize with empty unlabeled`
`data`
**for** $k \in \{1, \cdots, M\}$ `// Outer-loop with M`
`episodes`
**do**

  $\mathcal{T} \leftarrow \mathcal{D} \cup \hat{\mathcal{U}}$       `// Join updated`
  `pseudo-labels`
  **for** $(\mathbf{x}_i, \mathbf{y}_i) \in \mathcal{T}$   `// Inner-loop with K`
  `epochs`
  **do**

    $\theta_i = (\gamma_i, v_i, \alpha_i, \beta_i) = f^{(k-1)}(\mathbf{x}_i)$
    `// Evidental parameters`
    $\hat{\mathbf{y}}_i = \mathbb{E}[\mu] = \gamma_i$       `// Posterior`
    `prediction`
    $\mathcal{L} = \mathrm{L}(\hat{\mathbf{y}}_i, \mathbf{y}_i, \theta_i, \hat{\mathcal{W}}_i)$     `// Adaptive`
    `evidential loss via Eq. 5`
    $f^{(k-1)} = \mathrm{Update}(f^{(k-1)}, \mathcal{L})$
    `// Inner-loop update`

  **end**
  $f^{(k)} \leftarrow f^{(k-1)}$ `// Update teacher model`
  `for pseudo-labels`
  **for** $\mathbf{x}_i \in \mathcal{U}$ **do**

    $\hat{\theta}_i = (\hat{\gamma}_i, \hat{v}_i, \hat{\alpha}_i, \hat{\beta}_i) = f^{(k)}(\mathbf{x}_i)$
    $\hat{\mathbf{y}}_i = \hat{\gamma}_i$     `// Infer a new set of`
    `pseudo-labels`
    $\hat{\mathcal{U}}_i \leftarrow (\mathbf{x}_i, \hat{\mathbf{y}}_i)$       `// Update`
    `pseudo-labels`
    $\hat{\mathcal{W}}_i \leftarrow \mathrm{Var}[\mu]_i^{-1} = \hat{v}_i * (\hat{\alpha}_i - 1)/\hat{\beta}_i$
    `// Update adaptive weights`

  **end**

**end**

---

2004]. We select labels $y$ at the highest mode of probability distribution $\mathrm{p}_{\theta(t-1)}(y \mid \mathbf{x}_i)$, which corresponds to $y = \gamma_i^{t-1}$. We obtain the following approximate for the entropy:

$$\mathcal{H}(\mathcal{Y} \mid \mathcal{U}) \approx \mathcal{H}_{emp}(\mathcal{Y} \mid \mathcal{U}) = - \sum_{\mathbf{x}_i \in \mathcal{U}} \mathcal{E}_i^{t-1} \log \mathrm{p}_{\theta(t)}(\gamma_i^{t-1} | \mathbf{x}_i),$$
$$(11)$$

where the log probabilities are weighted by empirical probabilities as weights $\mathcal{E}_i^{t-1}$ evaluated at iterative step $t-1$ when plugged into Eq. 8 (also see Appendix. Eq. 3 for exact formula of Student's t-distribution)

$$\mathcal{E}_i^{t-1} = \mathrm{St}(y = \gamma_i, \sigma_{st,i}^2, 2\alpha_i) = \frac{1}{\sqrt{2\,\alpha_i\,\sigma_{st,i}^2}\,\mathrm{B}(\frac{1}{2}, \alpha_i)}.$$
$$(12)$$

To establish a relationship between empirical weights $\mathcal{E}_i^{t-1}$ and aleatoric $\mathbb{E}[\sigma_i^2]$ / epistemic $\mathrm{Var}[\mu_i]$ uncertainties we

rewrite

$$\sigma_{st,i}^2 = \frac{\alpha_i - 1}{\alpha_i} \left( \mathrm{Var}[\mu_i] + \mathbb{E}[\sigma_i^2] \right) \qquad (13)$$

$$\mathcal{E}_i^{t-1} = \frac{( \mathrm{Var}[\mu_i] + \mathbb{E}[\sigma_i^2] )^{-\frac{1}{2}}}{\sqrt{2}\,\mathrm{B}(\frac{1}{2}, \alpha_i)\sqrt{\alpha_i - 1}}. \qquad (14)$$

Empirical coefficients depend on aleatoric, epistemic uncertainties and $\alpha_i$ parameter, which can be interpreted as virtual observations in support of the variance estimation [Jordan, 2009]. In the limiting case $\alpha_i \gg 1$ one can approximate beta function via Stirling formula $\mathrm{B}(\frac{1}{2}, \alpha_i) \approx \sqrt{\pi}\,\alpha_i^{-\frac{1}{2}}$ and empirical weights become

$$\mathcal{E}_i^{t-1} \approx ( \mathrm{Var}[\mu_i] + \mathbb{E}[\sigma_i^2] )^{-\frac{1}{2}}. \qquad (15)$$

We can express the empirical coefficients depend both on aleatoric and epistemic uncertainties in a symmetric fashion.

We selected adaptive pseudo-labeling coefficients $\mathcal{W}_i$ in our pseudo-labeling approach Eq. 5 to be inverse epistemic/model uncertainties. We can see, that those coefficients directly relate to empirical coefficients derived from entropy minimization approach Eq. 14, as empirical coefficients also depend on model uncertainty in the inverse fashion. As the model uncertainty increases, the empirical coefficients $\mathcal{E}_i$ tend to decrease to minimize the entropy.

# 6 EXPERIMENTS

## 6.1 DATASET AND EXPERIMENTAL SETUPS

We evaluate PSEUD$\sigma$ using the QM9 dataset [Wu et al., 2018] under two settings[1]. *(A) Full-data*: We follow the previous works [Liu et al., 2021, Klicpera et al., 2020] where a 110,000/10,000/10,831 training/validation/testing split is obtained. We draw unlabeled data from PC9, a dataset of 99,234 molecules that consists of the same elements as QM9, curated by [Glavatskikh et al., 2019]. *(B) Low-data*: we set $k\%$ of QM9 full training set as the training set (i.e. $k\% \times 110,000$) and we remove the labels from the remaining (1-$k\%$) of the QM9 full training set and treat this as the unlabeled set. We evaluate PSEUD$\sigma$ for two $k$ values, 1 and 10 (meaning only 1,100/11,000 labelled QM data points are retained, respectively). A summary of the dataset statistics is presented in Table 1. Note that PC9 has a wider chemical diversity than QM9, demonstrated by wider distribution of distances of chemical bonds and more functional groups [Glavatskikh et al., 2019].

PSEUD$\sigma$ is model-agnostic. We evaluate it with two model backbones SchNet [Schütt et al., 2017] (PSEUD$\sigma$-S) and DimeNet++ [Klicpera et al., 2020] (PSEUD$\sigma$-D). We do not

---

[1]Code and datasets are available at `https://github.com/PfizerRD/pseudo`.

Table 1: Dataset statistics.

| Setting | Training Set | Validation Set | Testing Set | Unlabeled Set | OOD Set |
|---|---|---|---|---|---|
| Full-data | 110,000 (QM9) | 10,000 (QM9) | 10,831 (QM9) | 99,234 (PC9) | - |
| Low-data-1% | 1,100 (QM9) | 10,000 (QM9) | 10,831 (QM9) | 108,900 (QM9) | - |
| Low-data-10% | 11,000 (QM9) | 10,000 (QM9) | 10,831 (QM9) | 99,000 (QM9) | - |
| Out-of-distribution | 110,000 (QM9) | 10,000 (QM9) | 10,831 (QM9) | 99,234 (PC9) | 99,234 (PC9) |

experiment with the SOTA atomistic model SphereNet [Liu et al., 2021] because it is highly computationally expensive. Our result is conducted on two targets $\sigma_{\mathrm{HOMO}}, \sigma_{\mathrm{LUMO}}$, because the PC9 dataset only has these two targets. We use mean absolute error as the evaluation metric.

For baselines, we compare with 6 state-of-the-art baselines, including SchNet [Schütt et al., 2017], PhysNet [Unke and Meuwly, 2019], Cormorant [Anderson et al., 2019], MGCN [Lu et al., 2019], DimeNet++ [Klicpera et al., 2020], and SphereNet [Liu et al., 2021]. We report the best results taken from the original authors' paper while using the same fraction of data split in the full data setting. For PSEUD$\sigma$, we conduct two hyperparameter tunings on $\sigma_{\mathrm{HOMO}}$ with SchNet backbone on the validation MAE with full data/low-data setting, respectively. The optimal hyperparameter is then used for both targets. Note that the atomistic model itself has the same hyperparameters as used by the original authors.

## 6.2 RESULTS

**Overview of results.** We report performances of PSEUD$\sigma$ in full data (Table 2), low-data (Table 3), and out-of-distribution (Table 4) settings and find PSEUD$\sigma$ achieves the best performance across all settings, suggesting the robustness of the pseudo-labeling strategy. A systematic ablation study (Table 5) shows the importance of each module in PSEUD$\sigma$.

**PSEUD$\sigma$ improves on fully supervised QM calculations.** We compare PSEUD$\sigma$ against 6 state-of-the-art models in Table 2. PSEUD$\sigma$-D surpasses all baselines on both targets $\sigma_{\mathrm{HOMO}}, \sigma_{\mathrm{LUMO}}$. Notably, PSEUD$\sigma$-D improves the SOTA by 3.2 meV, a significant margin. Particularly, comparing PSEUD$\sigma$-S with SchNet and PSEUD$\sigma$-D with DimeNet++, we find PSEUD$\sigma$ can consistently improve even on the fully supervised setting by a large margin (8.1 meV for SchNet and 4.2 meV for DimeNet++), highlighting the utility of PSEUD$\sigma$ and the high quality of PC9 as unlabeled data. For PSEUD$\sigma$-D, we used the same data split and ran it three times and obtained a standard deviation of 0.432, which is still significantly better than the best baseline. Overall, the strong empirical result on full supervised setting shows that this direction of improving learning strategy instead of improving physics-based representation is promising.

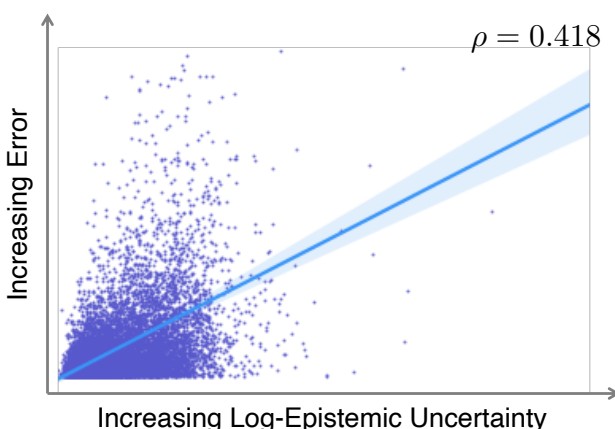

Figure 3: Uncertainty highly correlates to label quality.

**PSEUD$\sigma$ significantly improves on low-data QM calculations.** In Table 3, we investigate how PSEUD$\sigma$ can improve in the low-data regime with only 1% or 10% of the training data (i.e, using only 1,100 and 11,000 QM calculations). We observe PSEUD$\sigma$ can consistently and significantly improve prediction accuracy in $\sigma_{\mathrm{HOMO}}, \sigma_{\mathrm{LUMO}}$ across both low-data settings and both model backbones, suggesting PSEUD$\sigma$ can help prediction in realistic low-data settings (simulating the use of the more expensive QM levels of theory such as CCSD(T)/MP2). Notably, in $\sigma_{\mathrm{LUMO}}$ with 1% of QM9 data, PSEUD$\sigma$ improves upon SchNet by 57.8 meV, a considerable margin. We also observe that the gain margin is much more significant when the training dataset is smaller.

**PSEUD$\sigma$ improves out-of-distribution QM calculations.** Another realistic challenge is to infer accurately on unseen data distribution away from QM9. We conduct inference on the PC9 dataset (since the dataset already contains calculated $\sigma_{\mathrm{HOMO}}, \sigma_{\mathrm{LUMO}}$ values). We find PSEUD$\sigma$ can again significantly improve OOD accuracy over DimeNet++, a SOTA method, with over 16.0 meV improvement on $\sigma_{\mathrm{HOMO}}$ and 8.4 meV improvement on $\sigma_{\mathrm{LUMO}}$, highlighting the robustness of PSEUD$\sigma$.

**Evidential uncertainty highly correlates to label quality.** The motivation of PSEUD$\sigma$ to utilize uncertainty as a proxy of label quality is that they are highly correlated with each

Table 2: PSEUD$\sigma$ improves on full data setting. Reported metric is MAE. The lower the better.

| Property | Unit | SchNet | PhysNet | Cormorant | MGCN | DimeNet++ | SphereNet | PSEUD$\sigma$-S | PSEUD$\sigma$-D |
|---|---|---|---|---|---|---|---|---|---|
| $\epsilon_{\text{HOMO}}$ | meV | 41 | 32.9 | 36 | 42.1 | 24.6 | 23.6 | 32.9 | **20.4** |
| $\epsilon_{\text{LUMO}}$ | meV | 34 | 24.7 | 36 | 57.4 | 19.5 | 18.9 | 24.7 | **18.2** |

Table 3: PSEUD$\sigma$ improves on low-data regime. Reported metric is MAE. The lower the better.

| Low-Data Setting | | 1% QM9 (1,100) | | 10% QM9 (11,000) | |
|---|---|---|---|---|---|
| Property | Unit | SchNet $\to$ PSEUD$\sigma$ | DimeNet++ $\to$ PSEUD$\sigma$ | SchNet $\to$ PSEUD$\sigma$ | DimeNet++ $\to$ PSEUD$\sigma$ |
| $\epsilon_{\text{HOMO}}$ | meV | $265.4 \xrightarrow{+10.8} 276.2$ | $248.9 \xrightarrow{-18.7} 230.2$ | $119.0 \xrightarrow{-30.2} 88.8$ | $81.1 \xrightarrow{-13.7} 67.4$ |
| $\epsilon_{\text{LUMO}}$ | meV | $290.6 \xrightarrow{-57.8} 232.8$ | $229.3 \xrightarrow{-5.2} 224.1$ | $93.3 \xrightarrow{-15.0} 78.3$ | $60.8 \xrightarrow{-1.6} 59.2$ |

Table 4: Out-of-distribution best validation MAE.

| Property | Unit | SchNet | DimeNet++ | PSEUD$\sigma$-D |
|---|---|---|---|---|
| $\sigma_{\text{HOMO}}$ | meV | 243.4 | 230.4 | **214.4** |
| $\sigma_{\text{LUMO}}$ | meV | 225.0 | 184.2 | **175.8** |

Table 5: Ablation using SchNet as backbone on the fully supervised setting.

| Property | Unit | PSEUD$\sigma$-S | -pseudo-label | -uncertainty | -student | -uniform |
|---|---|---|---|---|---|---|
| $\epsilon_{\text{HOMO}}$ | meV | **32.9** | 38.9 | 47.7 | 41.4 | 37.2 |
| $\epsilon_{\text{LUMO}}$ | meV | **24.7** | 27.2 | 32.1 | 31.4 | 28.8 |

other for unseen molecules. In this experiment, we want to validate this hypothesis. We train on the complete QM9 training set with evidential uncertainty and then infer on the QM9 testing set. We find that the non-parametric Spearman correlation between MAE and epistemic uncertainty is 0.42 with a p-value $<$ 1e-16 (Figure 3). Additionally, we evaluate on PC9 out-of-distribution set, and the Spearman correlation is 0.35 with p-value $<$ 1e-16, suggesting our uncertainty is a robust measure of label quality.

**Ablations.** In Table 5, we conduct a systematic ablation study using SchNet as the backbone architecture on the fully supervised QM9 setting. We show that each component in PSEUD$\sigma$ is indispensable for PSEUD$\sigma$. In Table 2, we have reported original authors best performance following standard practices Klicpera et al. [2020], Liu et al. [2021]. To further clearly demonstrate the utility of pseudo-labeling, in -pseudo-label, we keep all hyperparameters the same but remove the pseudo-labeling part. We show that our pseudo-labeling strategy improves performance by a large margin. Next, in the -uncertainty ablation, we use a vanilla per-epoch pseudo-labeling strategy with no uncertainty. This ablation corresponds to the pseudo-labeling strategy in [Lee et al., 2013]. This decreases performances even as compared to the -pseudo-label strategy. Then, to compare against stan-

dard self-training, the -student ablation retrains a model in every episode as in [Xie et al., 2020] and we see decreased performance. From the last two ablations, we see that previous pseudo-labelling strategies have negative or limited transfer for QM while our approach achieved strong positive transfer. Lastly, the -uniform ablation uses the same weight for all pseudo-labels with no uncertainty reweighting. The decreased performance shows the importance of detection and adaptive removal of bad pseudo-labels, achieved by our evidential characterization of the molecular target property.

# 7 CONCLUSION

We introduce PSEUD$\sigma$, a simple, effective, model-agnostic pseudo-labeling strategy that can improve quantum calculations accuracy in abundant data, low data and out-of-distribution settings. PSEUD$\sigma$ learns from vast unlabeled data by assigning uncertainty-aware pseudo-labels. These pseudo-labels are adaptively selected to be absorbed into the model via an episodic schedule. Unlike earlier methods in QM that focuses on physics-based representation, we show the potential of this data-centric approach to improve performance on a task crucial to materials and therapeutic discovery.

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
