# OpenReview forum: "Uncertainty-Aware Pseudo-labeling for Quantum Calculations"
_auai.org/UAI/2022/Conference — UAI 2022 Poster_

### Official Review · Reviewer_cLmT · 2022-04-05

**Q2(1) Originality/Novelty:** 3
**Q2(2) Significance/Impact:** 3
**Q2(3) Correctness/Technical Quality:** 3
**Q2(6) Clarity Of Writing:** 3
**Q6 Overall Score:** 6
**Q8 Confidence In Your Score:** 4

**Q1 Summary And Contributions:**

The paper presents a pseudo-labeling approach for training e.g. message passing neural networks for molecular property prediction. Across two different NN architectures, the paper demonstrates benefits of including additional unlabelled data.


**Q2 Assessment Of The Paper:**

More detailed information regarding each of these aspects is given below:

**Q2(4) Quality Of Experiments (Optional):**

2: Fair: The experimental evaluation is weak: important baselines are missing, or the results do not adequately support the main claims.

**Q2(5) Reproducibility:**

2: Fair: Key resources (e.g., proofs, code, data) are unavailable but key details (e.g., proof sketches, experimental setup) are sufficiently well-described for an expert to confidently reproduce the main results.

**Q3 Main Strengths:**

Impressive results on homo/lumo prediction on QM9.
Well written and motivated.


**Q4 Main Weakness:**

The method is motivated somewhat ad hoc. The paper demonstrates that it works, but to me it does not clearly communicate in enough  detail how and why it works. I think the paper would be significantly stronger with a better theoretical foundation, perhaps backed up by minimal examples that clearly demonstrate the mechanism.

Code is not released (during the review period.)

In my view, the first 2.5 pages could be condensed to leave more room for e.g. simple examples demonstrating the principles behind the approach

**Q5 Detailed Comments To The Authors:**

I do not think you can state without reservation that SphereNet is the state of the art atomistic model. What about models such as PAINN and FCHL19? Maybe qualify the statement to say that is specifically for predicting homo and lumo on QM9.

It is my impression that both QM9 and PC9 datasets also include total energies. If that is correct, why did you not include energy in the experiments?


**Q7 Justification For Your Score:**

I think the results are significant and of broad interest, but the paper would be much stronger if it was more solidly based.


**Q9 Complying With Reviewing Instructions:**

1: Yes.

---

### Official Review · Reviewer_3RBm · 2022-04-09

**Q2(1) Originality/Novelty:** 2
**Q2(2) Significance/Impact:** 2
**Q2(3) Correctness/Technical Quality:** 3
**Q2(6) Clarity Of Writing:** 4
**Q6 Overall Score:** 6
**Q8 Confidence In Your Score:** 3

**Q1 Summary And Contributions:**

The paper proposes a pseudo-labeling strategy for semi-supervised learning that employs an uncertainty quantification to weight label proposals being incorporated in the learning process. This approach aims to solve the label-scarcity problem for predicting molecular quantum chemical properties. To this end, the authors suggest an adaptive weighting approach with connections to the widely-used entropy minimization, showing superior performance compared to recent QM prediction baselines.

**Q2 Assessment Of The Paper:**

More detailed information regarding each of these aspects is given below:

**Q2(4) Quality Of Experiments (Optional):**

2: Fair: The experimental evaluation is weak: important baselines are missing, or the results do not adequately support the main claims.

**Q2(5) Reproducibility:**

1: Poor: Key details (e.g., proof sketches, experimental setup) are incomplete/unclear, or key resources (e.g., proofs, code, data) are unavailable.

**Q3 Main Strengths:**

- Interesting connection between the entropy minimization and epistemic uncertainty minimization as proposed in the paper.
- Application to an interesting domain that seems challenging


**Q4 Main Weakness:**

- Lack of highly related pseudo-labeling related literature: See detailed comments.
- Insufficient consideration of baselines in evaluation: Pseud\sigma should be compared to other recent pseudo-labeling approaches that could “wrap” existing QM prediction models (just as Pseud\sigma does).
- Too little information for reproducibility and statistical significance: Many details are missing for reproduction, also more datasets could be considered.


**Q5 Detailed Comments To The Authors:**

Major:
- Lack of highly related pseudo-labeling related literature: The presented approach can be regarded as a somewhat generic pseudo-labeling strategy and should be put in relation to existing methods. E.g., for uncertainty awareness (Berthelot et al., 2019: “ReMixMatch: Semi-supervised learning with distribution alignment and augmentation anchoring” [+ others of that author and their variants], Lienen et al., NeurIPS 2021: “Credal Self-Supervised Learning”), specifically for adaptive weighting in semi-supervised learning (Ren et al., NeurIPS 2020: “Not all Unlabeled Data are Equal: Learning to Weight Data in Semi-supervised Learning”), and much more. Many of them are evaluated in image classification, but describe generic concepts, which could be also used for QM predictions.
- Insufficient consideration of baselines in evaluation: Basically, the approach is compared to non-semi-supervised methods, which is a somewhat distorted comparison due to different target availability. As this method can be seen as a somewhat generic pseudo-labeling approach, one should also compare it to different pseudo-labeling wrapper methods, such as the work in Rizve et al, 2017 or Lienen et al., 2021.
- Too little information for reproducibility and statistical significance: How many different seeds were considered? What about standard deviations of the results? How statistically significant are the results? What was the exact procedure to optimize the hyperparameters? Why only a single dataset (and the ood data)? …

Minor:
- Notation in problem formulation: D = {\mathcal{X}, \mathcal{Y}} (should be a subset of the cartesian product \mathcal{X} \times \mathcal{Y})
- The nature of the "episodic pseudo-labeling" is certainly not a novelty. Not updating the pseudo-labels in each mini-batch, but rather after each epoch, can be considered as the "basic self-training" paradigm (see literature before Lee, 2013). Recent methods could also make use of this delayed propagation, but, however, as the model uncertainty decreases, the model has incentives to learn from the latest, probably less uncertain and thus more reliable estimates. Practically speaking, this typically works better.
- Seeing different uncertainty quantifiers in a further ablation study would be interesting, e.g., Monte Carlo-sampling methods.


**Q7 Justification For Your Score:**

My main concern is the insufficient exploration of previous approaches in the field of pseudo-labeling, combined with the lack of a proper empirical evaluation to support all claims. This work is certainly not reproducible from just the paper and the supplementary material given.

**Q9 Complying With Reviewing Instructions:**

1: Yes.

---

### Official Review · Reviewer_RrzA · 2022-04-13

**Q2(1) Originality/Novelty:** 3
**Q2(2) Significance/Impact:** 2
**Q2(3) Correctness/Technical Quality:** 3
**Q2(6) Clarity Of Writing:** 2
**Q6 Overall Score:** 5
**Q8 Confidence In Your Score:** 3

**Q1 Summary And Contributions:**

This paper focuses on incorporating QM pseudo-labels to alleviate the fundamental label scarcity issue, which targets the challenges, including low-data and out-of-distribution generalization.

**Q2 Assessment Of The Paper:**

More detailed information regarding each of these aspects is given below:

**Q2(4) Quality Of Experiments (Optional):**

3: Good: The experimental evaluation is adequate, and the results convincingly support the main claims.

**Q2(5) Reproducibility:**

3: Good: Key resources (e.g., proofs, code, data) are available and key details (e.g., proofs, experimental setup) are sufficiently well-described for competent researchers to confidently reproduce the main results.

**Q3 Main Strengths:**

1)	Unlike previous QM methods, this paper investigates more cases of low-data and out-of-distribution settings;
2)	This paper focuses on data-centric approaches to learn from unlabeled molecules;
3)	This paper proposes a simple, effective, theoretically motivated pseudo-labeling strategy for QM to integrate episodic scheduling and downplay low-quality pseudo-labels informed by uncertainty;
4)	This paper shows that the proposed method can improve the QM accuracy in several settings.

**Q4 Main Weakness:**

There still exist some points to be improved for this paper. For example, Figure 3 is not explained in the main body of this paper. Table 5 should use the same format (with a bold borderline) as other tables. The name of the property in Table 2 should be consistent with the description in the main body. The author should be careful about these details.

**Q5 Detailed Comments To The Authors:**

There still exist some points to be improved.
Here are some major comments:
1)	In Table 5, as the ablation is conducted on SchNet, then it’s better to use the method name PSEUDO-S in the table.
2)	Can you add more statistical results about the proportion for different scales of the adaptive weight for the detected pseudo-labels?
3)	The explanation for Figure 3 cannot be found in this paper. It should add some comments for Figure 3.
4)	Table 5 should also have a bold border like other tables.
5)	In the body of this paper, it mentions “…surpasses all baselines on both targets …”
Note that the typing of the target name (Property) in Table 2 is not consistent with the description. Please be careful about spelling.

**Q7 Justification For Your Score:**

See Q5

**Q9 Complying With Reviewing Instructions:**

1: Yes.

---

### Official Review · Reviewer_R5pp · 2022-04-21

**Q2(1) Originality/Novelty:** 2
**Q2(2) Significance/Impact:** 2
**Q2(3) Correctness/Technical Quality:** 2
**Q2(6) Clarity Of Writing:** 4
**Q6 Overall Score:** 5
**Q8 Confidence In Your Score:** 3

**Q1 Summary And Contributions:**

This paper develops uncertainty-aware mechanism for pesudo-labeling in quantum calculation.

**Q2 Assessment Of The Paper:**

More detailed information regarding each of these aspects is given below:

**Q2(4) Quality Of Experiments (Optional):**

2: Fair: The experimental evaluation is weak: important baselines are missing, or the results do not adequately support the main claims.

**Q2(5) Reproducibility:**

3: Good: Key resources (e.g., proofs, code, data) are available and key details (e.g., proofs, experimental setup) are sufficiently well-described for competent researchers to confidently reproduce the main results.

**Q3 Main Strengths:**

1. well organized and clearly presented.
2. the idea is simple and straightforward and the experimental results support its claim.

**Q4 Main Weakness:**

1. in eq(5), there is no explicit trade-off hyperparameters between ground-truth labels and pseudo-labels, which might make the model incomplete. By adding a trade-off parameter, the control over how to use pseudo-labels will be more flexible.
2. Eq(4) itself is generating uncertainty and affect the learning of hyperparameters in priors. Thus, it is believed that even without using pseduo labels, a prior can be studied. It would be great if this can be used as a baseline in the experiments.


**Q5 Detailed Comments To The Authors:**

see above main strenghths and weakness.
A minor issue:
in section 3 problem formulation about the definition of {y_1,...,y_N}.
not clear here: N is the number of molecules, but {y_1,...,y_N} is defined as a set of quantum mechanical properties for "each" molecule. From my understanding, one molecule has one corresponding y_i instead of a set.

**Q7 Justification For Your Score:**

Overall, the paper is good. The main concern is as stated in the first main weakness.

**Q9 Complying With Reviewing Instructions:**

1: Yes.

---

### Decision · Program_Chairs · 2022-05-15

**Decision:**

Accept (Poster)

**Comment:**

Meta Review: This paper develops an uncertainty-aware mechanism for pseudo-labeling to alleviate the label scarcity issue in quantum calculation.  It receives four reviews: two weak accepts and two borderline accepts.  The reviewers in general believe the paper addresses a difficult problem, the proposed method is novel, and the experimental results are strong.  The main concerns include missing some technical details, missing some related work, nsufficient baseline, and weak theoretical foundation.  The authors effectively address most of these concerns during rebuttal.